# Transient non-Hermitian skin effect

Zhongming Gu[1], He Gao[2] ✉, Haoran Xue[3] ✉, Jensen Li[4], Zhongqing Su[2] & Jie Zhu[1] ✉

The discovery of non-Hermitian skin effect (NHSE) has opened an exciting direction for unveiling unusual physics and phenomena in non-Hermitian system. Despite notable theoretical breakthroughs, actual observation of NHSE's whole evolvement, however, relies mainly on gain medium to provide amplified mode. It typically impedes the development of simple, robust system. Here, we show that a passive system is fully capable of supporting the observation of the complete evolution picture of NHSE, without the need of any gain medium. With a simple lattice model and acoustic ring resonators, we use complex-frequency excitation to create virtual gain effect, and experimentally demonstrate that exact NHSE can persist in a totally passive system during a quasi-stationary stage. This results in the transient NHSE: passive construction of NHSE in a short time window. Despite the general energy decay, the localization character of skin modes can still be clearly witnessed and successfully exploited. Our findings unveil the importance of excitation in realizing NHSE and paves the way towards studying the peculiar features of non-Hermitian physics with diverse passive platforms.

Topological phases of matter have been widely studied in various platforms, including both electronic materials[1,2] and artificial structures[3–6]. Conventional topological band theory is built on the system's Hamiltonian, which is often assumed to be Hermitian. However, for a system coupled to the open environment or with gain and/ or loss modulation, a non-Hermitian Hamiltonian needs to be employed[7]. In contrast to the Hermitian case, a non-Hermitian Hamiltonian can host complex eigenvalues with nonorthogonal eigenmodes and has unique symmetries and bandgaps without Hermitian counterparts. These sharp differences lead to many surprises in topological physics and make non-Hermitian topology currently a hot topic under extensive research[8–10].

One paradigmatic topological phenomenon in non-Hermitian systems is NHSE, which refers to the localization of an extensive number of eigenmodes at the boundaries[11–16]. NHSE has profound impacts on band topology as it reflects a novel point-gap topology that is unique to non-Hermitian systems[13,17–19] and leads to a breakdown of the conventional bulk-boundary correspondence[14,15,20]. The boundary-localized eigenmodes in NHSE, namely the skin modes, only exist in

point-gapped systems that are intrinsically non-Hermitian and thus are fundamentally different from other boundary modes such as the Tamm states and topological edge states that can arise without the aid of non-Hermiticity. Due to its unconventional physical properties, NHSE has also been found to be useful in various applications, including wave funneling[21], enhanced sensing[22,23], and topological lasing[24,25].

NHSE has been demonstrated in a number of active platforms, such as optical fiber loops with optical intensity modulators[21], circuit lattices with negative impedance converters[26–28] and phononic crystals connected to external circuits[29–32]. The active components, while introducing gain and making NHSE easy to observe, largely enhance the complexities and instabilities of the systems. For example, when the lattice size is large and the excitation is far away from the boundary, the high intensity of the amplifying wavepacket can cause significant nonlinear effects[33]. On the other hand, NHSE can also exist in purely lossy systems[34]. Despite the advantages of being easy to construct and stable, one crucial obstacle in passive systems is the difficulty in exciting the skin modes[35,36]. Owing to the rapid decay of

[1]Institute of Acoustics, School of Physics Science and Engineering, Tongji University, 200092 Shanghai, China. [2]Department of Mechanical Engineering, The Hong Kong Polytechnic University, Hung Hom, Kowloon, Hong Kong SAR, China. [3]Division of Physics and Applied Physics, School of Physical and Mathematical Sciences, Nanyang Technological University, Singapore 637371, Singapore. [4]Department of Physics, The Hong Kong University of Science and Technology, Kowloon, Hong Kong, China. ✉e-mail: h.e.gao@connect.polyu.hk; haoran001@e.ntu.edu.sg; jiezhu@tongji.edu.cn

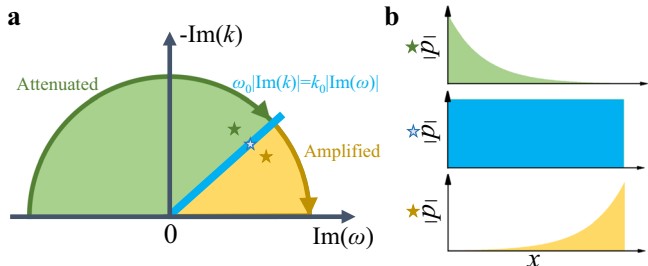

**Fig. 1 | Conceptual illustration of the virtual gain effect. a** The effects due to simultaneous modulations on $k$ and $\omega$ in the complex plane. The green and yellow regions exhibit virtual loss and gain effects, respectively. The blue line denotes the critical points. **b** The field distributions for the three cases that are marked as stars in (**a**).

energy, it is in general hard to observe NHSE, especially when the source is away from the boundary where the skin modes are localized. In such a case, instead of observing the localization of the skin modes, one observes that the excited field is localized around the source (see Fig. 2d).

In this work, we show that the above problem can be overcome using the concept of virtual gain, originally developed in parity-time symmetric systems[37–40]. This technique has also been applied to achieve virtual optical pulling force[41]. The key idea is that, under a complex-frequency excitation, the intensity can increase along the propagation direction despite the eigenmodes moving in the same direction being lossy, thus mimicking the gain effect. In a passive lattice with NHSE, such a method can be exploited to excite the skin modes with an arbitrary excitation position, and a unique NHSE, dubbed transient NHSE, can be observed in a quasi-stationary stage where the system's intensity drops in time but the localized profile persists. We provide an illustrative tight-binding model and a concrete acoustic crystal to demonstrate the feasibility of the idea. Furthermore, the acoustic model is realized experimentally and the virtual gain effect is verified through time-resolved field mapping.

## Results

### Virtual gain effect
The virtual gain effect is realized by so-called complex-frequency excitation[37,42]. Conventionally, we use the expression $e^{i(\omega t - kx)}$ to represent a propagating wave that travels along the $x$ direction with a frequency of $\omega$, while the $e^{i\omega t}$ is often omitted for simplicity in the case of monochromatic incidence. When gain and loss are considered as the characteristics of the materials, a complex wave number $k$ needs to be adopted: $k = k_0 n$, where $k_0$ is the wave number in the free space and $n = n_0 + in'$ is the complex refractive index of the materials. Then, there will have a non-propagation term $e^{k_0 n' x}$ that denotes the attenuation or amplification, depending on the sign of $n'$. For a complex-frequency excitation with $\omega = \omega_0 + i\omega'$, the intensity distribution shares a similar profile with that induced by the complex refractive index. When both gain/loss and the complex-frequency excitation are present, the amplification/attenuation of the wave is dependent on both factors, as illustrated in Fig. 1a. Since the system is passive, we only consider the imaginary part of $k$ varies in the negative axis. Specifically, when $(\frac{\omega'}{\omega_0} - n') > 0$, the intensity decays along the propagation direction, corresponding to the loss effect. On the contrary, when $(\frac{\omega'}{\omega_0} - n') < 0$, the intensity increases along the propagation direction, corresponding to the gain effect. Moreover, when $|\frac{\omega'}{\omega_0}| = |n'|$, the modulations in the time domain and space domain will generate a plane wave pattern without distortions. The field distributions of the above three cases are illustrated in Fig. 1b.

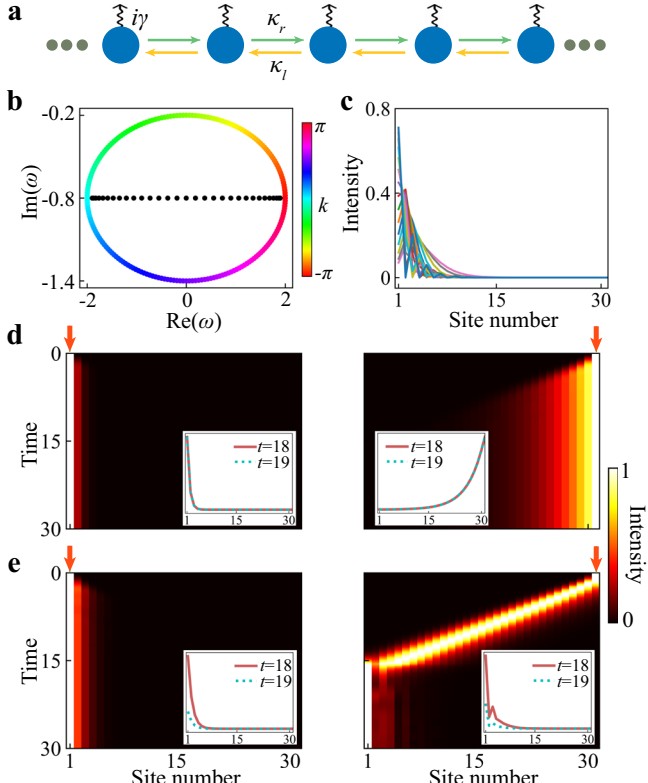

**Fig. 2 | Numerical results for the 1D tight-binding model. a** Schematic of the lattice, where $\gamma$ denotes the on-site loss and $\kappa_r (\kappa_l)$ are couplings from left to right (right to left). **b** The eigen spectra for the lattice shown in (**a**) under PBC (colored dots) and OBC (black dots), respectively. **c** Plot of all eigenmodes under OBC for a finite lattice with 31 sites. **d** Time evolutions of the intensity field in a finite lattice with a monochromatic excitation ($\omega = 0.5$) at the left end (left panel) and the right end (right panel). **e** Similar to (**d**) but for a complex-frequency excitation ($\omega = 0.5 + 0.6i$). **d**, **e** The intensity at each time slice is normalized by the maximum value at that moment for easy visualization and the insets show the signal strength without normalization at $t = 18$ and $t = 19$. The lattice parameters used in the calculations are $\kappa_r = 0.7$, $\kappa_l = 1.3$, and $\gamma = 0.8$.

### Tight-binding model
To illustrate the idea, we first consider a simple one-dimensional (1D) lattice model (see Fig. 2a):

$$H = \sum_j \left( \kappa_r c_{j+1}^\dagger c_j + \kappa_l c_j^\dagger c_{j+1} + i\gamma c_j^\dagger c_j \right), \tag{1}$$

where $\kappa_{l/r}$ are hopping strengths and $\gamma < 0$ is the on-site loss. This model is simply the minimal model for NHSE (i.e., the Hatano–Nelson model with nearest-neighbor asymmetric hoppings)[43] with additional on-site loss. Note that while here we use this specific model for illustration, our proposed method is applicable to various passive systems with NHSE, as evident in the following discussions and experiments. When $\gamma = 0$, this lattice is known to host NHSE induced by the asymmetric hoppings (i.e., $\kappa_r \neq \kappa_l$), with the eigen spectrum under periodic boundary condition (PBC) forms a closed loop in the complex plane and the eigenmodes under open boundary condition (OBC) are all skin modes. A nonzero $\gamma$ just shifts the eigenvalues of all modes along the imaginary axis while keeping the nontrivial point-gap topology unchanged. In our calculations, we tune $\gamma$ such that all modes are decaying modes (see Fig. 2b, c), which allows us to study the behaviors of a passive NHSE lattice under different excitations.

Consider an excitation of the form $e^{i\omega t}\psi_0$, where $\omega$ denotes the excitation frequency/energy (which can be complex), $t$ is time and $\psi_0$ is

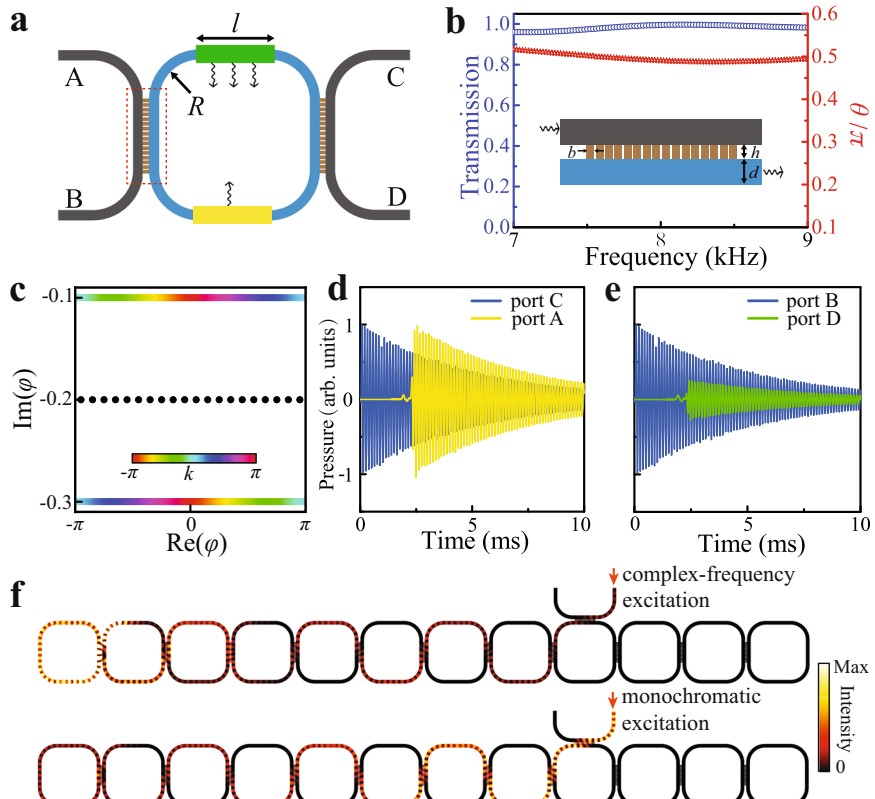

**Fig. 3 | Numerical results for the acoustic ring resonator lattice. a** The acoustic unit cell. The radius, $R$, of the circular part is 5.6 cm and the width, $l$, of the straight part is 9.6 cm. The green and yellow areas with arrows represent areas with different losses. The red dashed boxes denote the coupling regions. **b** Plots of $\theta$ (red stars) and the transmission coefficient (blue circles) between two neighboring rings against frequency. The inset shows the simulation geometry, which consists of an array of narrow waveguides (16 in total) that are used to couple the neighboring rings. The width and thickness of each narrow tube $h = 0.7$ cm and $b = 0.35$ cm, arrayed with a period of 0.47 cm. The width, $d$, of the ring resonator is 1.4 cm. **c** The eigen spectra for the PBC and OBC lattices are derived from the transfer matrix method with $\theta = 0.5\pi$. The colored dots and black dots denote the results for the PBC lattice and OBC lattice, respectively. **d, e** Simulated pressure profiles of the input and output signals with complex frequency for the waves propagate along a path with (**d**) less loss and (**e**) more loss, respectively. The transient simulations are conducted by finite element method. **f** Simulated field distributions at 15 ms after the lattice is injected with monochromatic excitation and complex-frequency excitation, respectively.

---

the excitation profile at $t = 0$. When $\omega$ is a real number, the excitation intensity remains unchanged in time, which is the usual case in experiments. Figure 2d shows the time evolution of the intensity field in a finite lattice when the excitations with $\omega = 0.5$ are located at the left and the right ends, respectively. As can be seen, the field is always localized around the position of the excitation due to the overall loss in the system, which fails to reveal the NHSE.

Next, we set $\omega$ to be a complex number, i.e., $\omega = \omega_0 + i\omega'$. The parameter $\omega'$ controls the amplification or decay of the excitation in time. For an excitation with complex frequency $\omega$, at a fixed instant, the intensity (normalized to the input) will increase along the propagation direction when the decay in excitation is faster than the actual decay of the mode in the lattice at the corresponding frequency. To see the effects of such a virtual gain, we again calculated the intensity evolution using the same lattice as in Fig. 2d but with a complex-frequency excitation ($\omega' = 0.6$). As shown in Fig. 2e, now the NHSE can be observed, even when the excitation position is far away from the boundary where skin modes are localized. We note that, in such a passive system with a complex-frequency excitation, the total intensity will decay in time. However, during the time window when the quasi-stationary profile is developed and the system's intensity is higher than the background noise, the transient NHSE can always be observed (see the insets in Fig. 2e). Apart from the localization property, more subtle features of NHSE, such as the non-local response, can also be captured with the complex-frequency excitation, as provided in the Supplementary Note 2 of Supplementary Materials.

## Acoustic model

We now apply the above scheme to a realistic acoustic ring resonator lattice with only lossy elements. As depicted in Fig. 3a, each unit cell of this lattice contains a site ring (depicted in gray) and a link ring (depicted in blue) of identical sizes. Both rings are hollow, filled with air, and surrounded by hard walls. The upper and lower parts of the link ring have different losses (see Fig. 3a), which induces NHSE[44,45]. In this lattice, there are two sets of modes with opposite circulation directions in the ring resonators, which can be made to be decoupled by carefully designing the structures to minimize the backscattering. Thereafter, we focus on the modes with clockwise/counterclockwise circulation direction in the link/site ring.

We use the transfer matrix method to calculate the eigen spectra under PBC and OBC[46–48]. In each coupling region (denoted by the red dashed boxes in Fig. 3a), the incoming and outcoming waves are related by a scattering matrix:

$$S = \begin{bmatrix} \cos\theta & -i\sin\theta \\ -i\sin\theta & \cos\theta \end{bmatrix}. \tag{2}$$

where $\theta$ describes the coupling strength. Since the loss is negligible in the coupling region, the $S$ matrix is unitary. Numerically, the value of $\theta$ can be extracted from a two-port transmission simulation. The coupling regions are carefully designed such that $\theta$ in a broad frequency window ranging from 7000 to 9000 Hz is around $0.5\pi$, corresponding to the perfect transmission case (see Fig. 3b). More

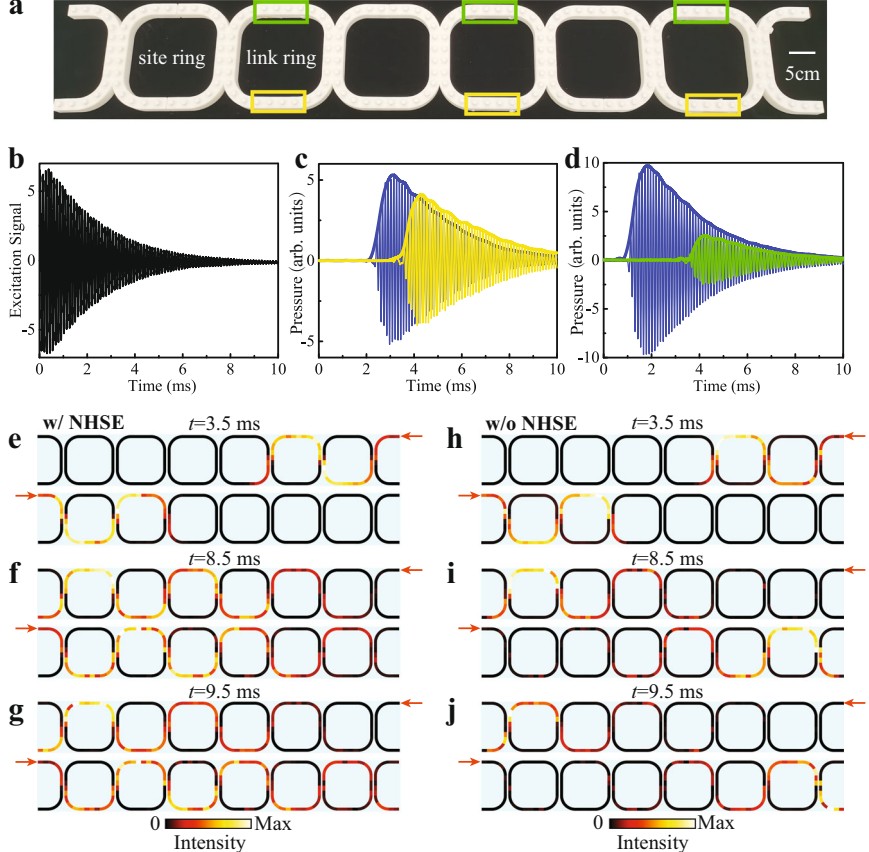

**Fig. 4 | Experimental demonstration in a 1D acoustic lattice. a** Photo of the 3D-printed experimental sample, consisting of six ring resonators and two half rings on the boundaries. The green and yellow boxes indicate the areas with additional loss. **b** The measured electric signal imposed on the loudspeaker. **c** The time evolution of sound pressure for the left-moving acoustic wave. The signals are measured at the middle position of the first site ring (blue) and second site ring (yellow), respectively. **d** The time evolution of sound pressure for the right-moving acoustic wave. The signals are measured at the middle position of the first site ring (blue) and second site ring (green), respectively. **e**–**g** Acoustic energy fields measured at (**e**) 3.5 ms, (**f**) 8.5 ms, and (**g**) 9.5 ms for both right and left incidences. **h**–**j** Acoustic energy fields measured in an empty waveguide without additional loss under the same complex-frequency excitation at (**h**) 3.5 ms, (**i**) 8.5 ms and (**j**) 9.5 ms for both right and left incidences.

details about the design procedure can be found in the Supplementary Note 3 of Supplementary Materials. Given the boundary condition, the scattering equations can be cast into a Floquet eigenproblem, with the round trip phase $\varphi$ in each ring playing the role of quasienergy (see Supplementary Note 1 for more details).

Figure 3c shows the eigenfrequencies for both PBC (colored dots) and OBC (black dots) when $\theta = 0.5\pi$ (the derivation can be found in Supplementary Note 1 of Supplementary Materials). The PBC spectrum features two bands with opposite group velocity and distinct imaginary parts, which leads to the formation of skin modes under OBC. It is worthwhile to note that the PBC spectrum does not form a closed loop in the complex plane and the OBC spectrum spans the whole quasienergy range, which are unique features of anomalous NHSE in Floquet systems[35]. Upon reducing the coupling strength $\theta$, the PBC spectrum will evolve into a closed loop similar to the one given in Fig. 2b (see Supplementary Note 4).

To demonstrate how NHSE can be observed in this all-passive acoustic lattice, we first show the generation of virtual gain in a single-cell setup as shown in Fig. 3a. The right-moving acoustic waves will propagate from port B to port D through the green area, whereas the left-moving waves will propagate from port C to port A through the yellow area. As discussed above, the effect of the complex-frequency excitation depends on the relative speed of attenuation caused by the system loss and the excitation decay. Thus, if the decay of the excitation is larger than the decay caused by the yellow area but smaller than the decay caused by the green area, the left-moving waves will have

virtual gain while the right-moving waves are still decaying. Such a scenario is numerically demonstrated in Fig. 3d, e, where blue curves are the input signals and yellow/green curves are output signals. It is clear that at a fixed time after the signal reaches the output ports, the signal at port A/D is larger/smaller than the input one, thereby realizing virtual gain/loss for left/right-moving waves. With this designed complex-frequency excitation, the NHSE can be clearly visualized, as shown in the top panel of Fig. 3f. At 15 ms after the excitation, the skin mode at the left boundary has been well established. For comparison, we also study the field distribution at the same time under the monochromatic excitation, as shown in the bottom panel of Fig. 3f. It is obvious that the acoustic energy has been blocked at the input position and decays along the propagation direction since the system is purely dissipative.

## Experiments

We fabricate a 1D acoustic lattice with six ring resonators to realize the proposed scheme experimentally (Fig. 4a). There are two half rings attached at both ends of the lattice to inject sound signals. In addition to the background loss, different amounts of additional loss are introduced to the areas denoted by the green and yellow boxes in Fig. 4a. In the experiments, an input signal with a complex frequency of $8250 (1 + 0.05i)$ Hz is launched from one excitation port and a 1/4-inch microphone is successively inserted into the small holes in the whole structure to probe the acoustic waves. The measured time-varying electric signal imposed on the loudspeaker is shown in Fig. 4b. By

calculating the enveloping line of the temporal signals, we can obtain the time-resolved amplitude of the sound waves in the whole lattice. More experimental details can be found in "Methods" and Supplementary Note 5.

We first adjust the amounts of absorbing materials inserted into the yellow and green areas, such that sound propagating through the yellow/green area exhibits virtual gain/loss. This can be verified by measuring the sound pressure at the adjacent site rings. As shown in Fig. 4c, for left-moving transmission, the pressure amplitude at the second site ring (the yellow curve) is higher than that at the first site ring (the blue curve) at the same instant of time after traveling through the yellow area, thus realizing the virtual gain effect. By contrast, the situation is reversed for the right-moving transmission (see Fig. 4d).

Next, we probe the signatures of the NHSE. Figure 4e–g shows the acoustic intensity fields measured at three different times for both right and left incidences. At $t = 3.5$ ms, the sound waves have just passed through two rings and the quasi-stationary stage has not been reached yet (Fig. 4e). At $t = 8.5$ ms, the sound waves have gone across the whole lattice and it is clearly seen that the left-moving wave is amplifying while the right-moving wave is decaying (Fig. 4f), which signifies the NHSE. Moreover, at $t = 9.5$ ms, the measured intensity distribution, after normalization to its maximum value, is almost the same as the one at $t = 8.5$ ms (Fig. 4g). This indicates the quasi-stationary stage can remain stable for a finite time window before the signal decays to the noise level. The detailed time evolutions of the intensity field appear as movies, which are provided in Supplementary Note 6 of Supplementary Materials. As a comparison, we also measured the time-resolved acoustic intensity fields for an acoustic lattice without addtional loss (i.e., no absrobing materials are inserted into the yellow and green regions). In this case, the same complex-frequency excitation always produces localization at the opposite end to the excitation (see Fig. 4h–j), in contrast to the lattice with NHSE where the localization position is independent of the excitation position. It should be emphasized that the NHSE can be observed in a broad bandwidth with the high coupling strength, although we just use one specific frequency for the demonstration.

## Discussion

The complex-frequency excitation approach, as we show in this work, provides a universal route to study NHSE in passive platforms. Our work that demonstrates the feasibility of visualizing virtual gain and skin mode localization is the first step towards this direction. Many other interesting phenomena associated with NHSE, such as critical NHSE[49], NHSE with disorder[50] and self-healing of skin modes[51], can be studied using complex-frequency excitation in future studies.

While our demonstration is in acoustics, similar experiments can also be done in various passive systems where time-dependent excitation and time-resolved measurements can be done, such as electric and microwave systems. Since our acoustic system has a relatively large background loss due to the narrow regions in the structures, we anticipate other platforms with smaller intrinsic loss would have larger time windows for the transient NHSE.

## Methods
### Numerical simulations
All the simulations were performed with COMSOL Multiphysics, pressure acoustics module. The density and sound speed for the background medium, air, are set to be 1.21 kg/m$^3$ and 343 m/s, respectively. All the boundaries are considered as acoustically hard boundaries. When calculating the transmission and coupling angle (Fig. 3b), the simulation is conducted in the frequency domain. As illustrated in the inset in Fig. 3b, the coupling angle is defined as $|\theta| = \text{atan}(|p_{LB}|/|p_{LT}|)$, where the subscripts $LT$ and $LB$ represent the acoustic wave transmitted from left input, with incident signal $p_i$, to top output and bottom output at the right boundary, respectively.

When $p_{LT}/p_i > 0$($p_{LT}/p_i < 0$), the coupling angle can be obtained directly as $\theta = |\theta|(\theta = \pi - |\theta|)$. The transmission coefficient is the ratio of $|p_{LB}|^2$ to $|p_i|^2$. When calculating the temporal signals and field distributions in the sample (Fig. 3d–f), the simulations are conducted in the transient domain with a resolution of 1 μs. The complex-frequency excitation is set to be $8250 + 200i$ Hz.

### Experimental details
The sample is fabricated via 3D printing technology with a resolution of 0.1 mm. Arrayed small holes with a spacing of 2.8 cm are drilled at the top panel of the whole sample, allowing for signal detection. When the position is not probed, a stopper is inserted to the hole for acoustic sealing. An additional background loss was first introduced, and then the losses in the yellow/green boxes of Fig. 4a were decreased/increased to construct the virtual gain/loss effect. During the measurement, one port at the two ends of the structure is used for sound excitation, the other ports are sealed with absorptive materials to avoid backscattering. The sound signal is excited circularly, and each position is probed successively with the same recording length for post-processing.

The complex-frequency excitation is generated by employing a time-varying sinusoidal signal. The real part of the complex excitation is 8250 Hz, which is at the center of working frequency window of the acoustic lattice. To determine the imaginary part of the excitation, we conduct the measurement shown in Fig. 4c, d with the imaginary part of the excitation gradually increased from zero. When virtual gain and loss effects are respectively realized in the lower and upper parts of the link ring, the suitable value for the imaginary part of the excitation is then identified, which is 412.5 Hz. Finally, we prepare the time-dependent signal in advance, then sent it to the programmable signal generator to trigger the sound signal from the speaker.

## Data availability
The data that support the findings of this study are available from the corresponding author upon reasonable request.

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

## Acknowledgements
This work is supported by the Fundamental Research Funds for the Central Universities (Grant No. 22120220237) and by the Research Grants Council of Hong Kong SAR (Grant Nos. AoE/P-502/20 and 15205219).

## Author contributions
Z.G., H.G., H.X., and J.Z. conceived the idea. H.G., H.X., and J.Z. supervised the project. Z.G. designed the samples and performed the simulations. Z.G., H.G., and H.X. conducted the experiments and analyzed the data. Z.G., H.G., H.X., and J.Z. wrote the manuscript with inputs from J.L. and Z.S. All authors contributed to discussions of the manuscript.

## Competing interests
The authors declare no competing interests.
