## [Peer review file · Nature Communications]

REVIEWER COMMENTS

Reviewer #1 (Remarks to the Author):

Non-Hermitian skin effect (NHSE) typically requires gain media to be actually observed. In this manuscript, the authors apply the newly developed concept of virtual gain to experimentally implement NHSE in a passive platform. This interesting and timely work can inspire further studies of the virtual gain effect in other branches of non-Hermitian physics. In addition, the manuscript is well written and organized. Thus, in my opinion, this work meets the criteria of a publication in nature communications. Below, I have several comments.

1. As shown in fig. 1(b), the virtual gain effect can induce increased pressure when moving further away from the source. In the case of the coexistence scenario of the NHSE where the eigenmodes localized at the opposite boundary of the source and the virtual gain effects due to the complex-frequency excitations, how do the authors differentiate these two different effects in their demonstration? In other words, how do we make sure that the localized intensity at the opposite boundary (See Fig. 2e) is due to the NHSE instead of the virtual gain effect itself? Maybe a similar structure without NHSE needs to be considered as a reference for demonstrating the pure virtual gain effects.

2. The acoustic experiment is nice. But the reader might not be easy to understand the chosen specific parameters for the demonstration of the transient NHSE, especially in the situation where the identification of the acoustic model with the previous tight-binding model is only plausible. Could the authors present more details about the procedure for determining the chosen systems parameters and the specific complex excitation frequency?

Reviewer #2 (Remarks to the Author):

Review report to the manuscript "Transient non-Hermitian skin effect"

This work studies the non-Hermitian skin effect (NHSE) in a passive acoustic system. The authors show that although the NHSE can be realized in passive systems, the observations of the skin modes can be sometimes elusive if the excitation is far away from the localization boundary. They propose that a complex-frequency excitation technique can tackle this problem and allow observation of skin modes from far-away excitations. This technique is adopted from some published works dedicated to realize a virtual gain in the time dimension. With the virtual gain, the dissipated energy is re-gained during certain

time period, in which the skin modes can be observed. These results are experimentally verified in an acoustic ring-resonator design.

From the scientific point of view, the complex-frequency excitation technique adopted here is indeed an interesting way to visualize the rightward attenuation and leftward amplification, a signature of the NHSE. However, I find this technique is a bit pointless, mostly because the authors intentionally include two areas of excessive dissipation (the green and yellow regions in Fig. 3a) to increase the background dissipation. In fact, as long as the green region is dissipative, the condition for the NHSE, i.e., the asymmetric coupling, is satisfied. This means the yellow region can be lossless. In this way, the skin modes can be always observed, as shown in one of the authors' preprints (Ref. [45]), which reports the realization of NHSE using a similar design with loss-lossless treatment. In addition, the manuscript is poorly prepared, which is not scientifically sound for some parts and also lacks some very crucial technical details. For example, the authors use a HN model in the beginning to illustrate their theory. But the acoustic design later on has a very different property from the HN model (e.g., see the discrepancy between the colored lines in Figs. 2b and 3c). Aren't they supposed to justify each other? For the missing technical details, how to design the acoustic ring resonators to obtain a near perfect coupling and hence the decoupled clockwise and anticlockwise modes over a large frequency regime is never discussed. This is very important, because otherwise coupled modes would ruin the condition for NHSE. How to experimentally implement a complex frequency, which to my knowledge is very counterintuitive, is not discussed either. The descriptions of the effect of virtual gain are not clear. For example, sentences like "For a propagating mode with frequency ...the actual decay of the mode" are very confusing.

With these reasons, I do not recommend the manuscript for publication in Nature Communications.

Reviewer #3 (Remarks to the Author):

In the manuscript "Transient non-Hermitian skin effect" by Zhongming Gu, et al., the authors utilize the so-called virtual gain to reveal the non-Hermitian skin effect (NHSE). The NHSE effect is possible in non-Hermitian 1D or 2D systems and consists of the localization of many modes at edges. In non-Hermitian passive systems, the NHSE effect is hidden in the complex frequency plane. Authors show that exponentially decaying signals (frequency with complex imaginary part) enable excitation of these localized modes. The approach is discussed using the general toy tight-binding model, analyzed numerically in the acoustic domain and realized experimentally. The reported results of the work are rather curious and inspiring and can be useful for a broad range of applications. Overall, I believe that the paper is worth publication in Nature Communication after addressing the following comments:

1. It would be instructive to discuss the NHSE effect more and, in particular, how it differs from other edge modes (for example, photonic Tamm states) and why one needs gain to achieve this effect at the real frequency.

2. The model in Fig 3(a) comprises loss in the coupling, whereas the model in Fig 2 implies that the hopping strengths are real-valued. How this difference affects the results?
3. Please explain in the text how without the γ term, the eigenvalues of (1) under periodic boundary conditions (PBC) form a closed loop in the complex plane? I guess the Hamiltonian, in this case, is Hermitian and non-hermicity comes from the boundary conditions.
4. Page 2: "the intensity will increase along the propagation direction when the decay in excitation is faster than the actual decay of the mode." Actually, not intensity, but the ratio of it to the incoming signal would increase in time.
5. What is ϕ in Fig 3(c)? Does it relate to the frequency?
6. Please explain why the coupling regions' scattering matrix (2) is unitary, given that the couplers are lossy?
7. The concept of virtual gain was also proposed in the concept of optical force control [S Lepeshov, A Krasnok, Virtual optical pulling force, Optica 7 (8), 1024-1030 (2020)]

Response Letter to Reviewers

We appreciate the reviewers' helpful comments on our manuscript (NCOMMS-22-26029-T). In the letter below, we mark the quoted comments in *blue italics*, followed by our response. The detailed revisions are highlighted in **red**.

Reviewer #1:

Reviewer comments:

Non-Hermitian skin effect (NHSE) typically requires gain media to be actually observed. In this manuscript, the authors apply the newly developed concept of virtual gain to experimentally implement NHSE in a passive platform. This interesting and timely work can inspire further studies of the virtual gain effect in other branches of non-Hermitian physics. In addition, the manuscript is well written and organized. Thus, in my opinion, this work meets the criteria of a publication in nature communications. Below, I have several comments.

Response:

We would like to thank the reviewer for the encouraging remarks. We have carefully read the comments and revised the manuscript accordingly.

Reviewer comments:

1. As shown in fig. 1(b), the virtual gain effect can induce increased pressure when moving further away from the source. In the case of the coexistence scenario of the NHSE where the eigenmodes localized at the opposite boundary of the source and the virtual gain effects due to the complex-frequency excitations, how do the authors differentiate these two different effects in their demonstration? In other words, how do we make sure that the localized intensity at the opposite boundary (See Fig. 2e) is due to the NHSE instead of the virtual gain effect itself? Maybe a similar structure without NHSE needs to be considered as a reference for demonstrating the pure virtual gain effects.

Response:

We thank the reviewer for this insightful question and suggestion. We agree that the virtual gain itself can lead to increased intensity as the wave propagates away from the source. This effect is similar to the one induced by the NHSE when the source and the skin modes' are located at opposite boundaries. However, a key difference is, in systems with NHSE, localization always happens at the position where the skin modes localize. While for systems without NHSE, the localization induced solely by the virtual gain effect always happen at the opposite direction of the source. Therefore, as suggested by the reviewer, to confirm that the localization is indeed due to NHSE, we conducted additional experiments to compare the field distributions of current system with a similar structure without NHSE. We have added the new results to the new manuscript, and revised the main text at line 255 and Fig. 4 accordingly, which reads **“As a comparison, we also measured the time-resolved acoustic intensity fields for an acoustic lattice without additional loss (i.e., no absorbing materials are inserted into the yellow and green regions). In this case, the same complex frequency excitation always produces localization at the opposite end to the excitation [see Fig. 4h-j], in contrast to the lattice with NHSE where the localization position is independent of the excitation position.”**

Fig. 4 Experimental demonstration in a 1D acoustic lattice. **a**, Photo of the 3D-printed experimental sample, consisting of six ring resonators and two half rings on the boundaries. The green and yellow boxes indicate the areas with additional loss. **b**, The measured electric signal imposed on the loudspeaker. **c**, The time evolution of sound pressure for the left-moving acoustic wave. The signals are measured at the middle position of the first site ring (blue) and second site ring (yellow), respectively. **d**, The time evolution of sound pressure for the right-moving acoustic wave. The signals are measured at the middle position of the first site ring (blue) and second site ring (green), respectively. **e-g**, Acoustic energy fields measured at (e) 3.5 ms, (f) 8.5 ms and (g) 9.5 ms for both right and left incidences. **h-j**, Acoustic energy fields measured in an empty waveguide without additional loss under the same complex frequency excitation at (h) 3.5 ms, (i) 8.5 ms and (j) 9.5 ms for both right and left incidences.

Reviewer comments:

2. The acoustic experiment is nice. But the reader might not be easy to understand the chosen specific parameters for the demonstration of the transient NHSE, especially in the situation where the identification of the acoustic model with the previous tight-binding model is only plausible. Could the authors present more details about the procedure for determining the chosen systems parameters and the specific complex excitation frequency?

Response:

We thank the reviewer for this very helpful advice. In response, we have expanded the main text and Supplementary Information to include more details regarding the acoustic model and experiment design process.

To begin, the key to determining the system parameters is the design of the coupling region structure. An optimal coupling region should enable high transmission (to better visualize the NHSE) and low reflection (to avoid hybridization between the clockwise and counterclockwise modes). A coupling region of this type, the structural parameters are obtained through numerical optimization using full-wave simulation in COMSOL. In the revised Supplementary Information, we have added a new section (Note 3) to illustrate the design procedure, which reads:

There are two main considerations in designing the acoustic structure. The first one is that we want to achieve perfect coupling (i.e., full transmission) between two coupled ring resonators. This is for the purpose of clear visualization of the NHSE (the propagation path can be easily visualized in the perfect coupling case). The second consideration is the minimization of the reflection during the coupling process. This is crucial since our theory is based on the assumption that the clockwise and counterclockwise modes are decoupled.

To obtain optimal structural parameters that fulfill above requirements, we consider the structure shown in the top panel of Fig. S4a, which consists of two acoustic waveguides coupled by small tubes with fixed spacings (0.47 cm) and widths (0.7 cm). First, we sweep the thickness of the tubes (denoted by b) and frequency for a structure with 16 tubes to determine the value of parameter b and working frequency range with near zero reflection. According to the numerical results shown in the bottom panel of Fig. S4b, we adopt $b=0.35$ cm and working frequency around 8250 Hz (denoted by the grey star in Fig. S4a). Then, the coupling strength can be varied by changing the number of tubes in the coupling region. As shown in Fig. S4b, 16 tubes achieve perfect coupling in the frequency range of our interest, compared to the cases with 4 tubes and 10 tubes.

To check the performance of the design, we numerically calculated the pressure field at 8250 Hz of a basic unit cell to distinguish the modes with different circulations. As shown in Fig. S5, the mode injected from lower left side can be perfectly transmitted to the lower right side, without any noticeable reflection.

Fig. S4 The scattering properties of the coupling region. a The reflection property as a function of the structural parameter b . **b** The coupling angle with different numbers of tubes: 4 tubes, 10 tubes and 16 tubes.

Fig. S5 Acoustic wave propagation through one unit cell.

Second, the value of the complex frequency excitation is determined through the following procedure. The real part of excitation is chosen based on the working frequency. In our experiment, we pick 8250 Hz, which is at the center of the working frequency window. The imaginary part of excitation is determined experimentally. Since the upper and lower parts of the link ring have different extra losses, a value for the imaginary part of the excitation frequency must be chosen, so that acoustic waves going through the upper and lower parts exhibit virtual loss and gain, respectively. To determine this value, we gradually increase the imaginary part of the excitation and measure the output signals. When virtual gain is attained in the lower part while lossy transport remains in the upper part (i.e., the spectra shown in Fig. 4c-d in the main text), the corresponding excitation frequency is chosen. We have incorporated the discussions in the revised Methods part, which reads:

The complex-frequency excitation is generated by employing a time-varying sinusoidal signal. The real part of the complex excitation is 8250 Hz, which is at the center of working frequency window of the acoustic lattice. To determine the imaginary part of the excitation, we conduct the measurement shown in Fig. 4c-d with the imaginary part of the excitation gradually increased from zero. When virtual gain and loss effects are respectively realized in the lower and upper parts of the link ring, the suitable value for the imaginary part of the excitation is then identified, which is 412.5 Hz. Finally, we prepare the time-dependent signal in advance, then sent it to the programmable signal generator to trigger the sound signal from the speaker.

Reviewer #2

Reviewer comments:

Review report to the manuscript “Transient non-Hermitian skin effect”

This work studies the non-Hermitian skin effect (NHSE) in a passive acoustic system. The authors show that although the NHSE can be realized in passive systems, the observations of the skin modes can be sometimes elusive if the excitation is far away from the localization boundary. They propose that a complex-frequency excitation technique can tackle this problem and allow observation of skin modes from far-away excitations. This technique is adopted from some published works dedicated to realize a virtual gain in the time dimension. With the virtual gain, the dissipated energy is re-gained during certain time period, in which the skin modes can be observed. These results are experimentally verified in an acoustic ring-resonator design.

Response:

We would like to thank the reviewer for the time and efforts.

Reviewer comments:

From the scientific point of view, the complex-frequency excitation technique adopted here is indeed an interesting way to visualize the rightward attenuation and leftward amplification, a signature of the NHSE. However, I find this technique is a bit pointless, mostly because the authors intentionally include two areas of excessive dissipation (the green and yellow regions in Fig. 3a) to increase the background dissipation. In fact, as long as the green region is dissipative, the condition for the NHSE, i.e., the asymmetric coupling, is satisfied. This means the yellow region can be lossless. In this way, the skin modes can be always observed, as shown in one of the authors' preprints (Ref. [45]), which reports the realization of NHSE using a similar design with loss-lossless treatment.

Response:

This is an important subject that we would like to further elaborate on. In fact, it is the limitation of the work in Ref. [45] (Ref. [35] in the revised version) that motivates us to develop the new study presented in this paper.

First, we would like to point out that the experiment in Ref. [45] does not directly show NHSE but only asymmetric transmission. And Ref. [45] did not claim direct observation of NHSE but rather the anomalous Floquet variant. It is due to the fact that skin modes cannot be excited when the source is located away from the boundary where the skin modes localize. Such a challenge is well mirrored in the experimental results shown in Fig. 3(f) of Ref. [45], where the skin modes are not excited when the source is at the opposite end. The contrast in Figs. 2d and 2e in this manuscript further demonstrate how challenging it is to observe NHSE in passive systems without applying complex frequency excitation. As a result, the study in Ref. [45] does not contradict this work but rather highlight the importance of complex frequency excitation.

We are aware of a recent preprint (arXiv: 2207.09014 (2022)) by Prof. Henning Schomerus that was posted after our submission. It also theoretically highlighted the difficulty of observing NHSE in passive systems. Therefore, we believe that employing complex frequency excitation in NHSE study is both important and timely. In the revised manuscript, we have cited the above mentioned two papers in the introduction part. They are now Refs. [35] and [36].

Reviewer comments:

In addition, the manuscript is poorly prepared, which is not scientifically sound for some parts and also lacks some very crucial technical details. For example, the authors use a HN model in the beginning to illustrate their theory. But the acoustic design later on has a very different property from the HN model (e.g., see the discrepancy between the colored lines in Figs. 2b and 3c). Aren't they supposed to justify each other?

Response:

Although the HN model and the ring resonator lattice model can have somewhat different spectra, they are both 1D lattices supporting NHSE. One reason for starting with the HN model is because we want to introduce our concept to the readers using the most typical NHSE model. Besides, we wish to show that our strategy is general for any models with NHSE. The discrepancy in their spectra will not affect our demonstration. We can certainly make them look comparable by reducing the coupling strength θ in the ring resonator lattice model, as seen in Fig. S6 in revised Supplementary Information.

From the reviewer's comment, we realize that using two different models for theory and demonstration may confuse the readers who expect two identical models. To avoid this misunderstanding, we have made the following changes. At line 99, we have added a sentence to emphasize that our method is not model-specific: "Note that while here we use this specific model for illustration, our proposed method is applicable to various passive systems with NHSE, as evident in the following discussions and experiment". At line 180, we have added the following sentence: "Upon reducing the coupling strength θ , the PBC spectrum will evolve into a closed loop similar to the one given in Fig. 2b (see Supplementary Note 4)." Moreover, we have created a new section (Note 4) in Supplementary Information to show the spectra evolution of the ring resonator lattice model, which reads:

In this section, we show the PBC and OBC spectra of the ring resonator lattices for various values of the coupling strength θ . In the acoustic design, we optimize the structural parameters to obtain the perfect coupling (i.e., $\theta=0.5\pi$) for a better visualization of NHSE, which also gives rise to the straight lineshape in the PBC spectrum that is distinct from the closed loop spectrum in conventional NHSE (for example, see Fig. 2b in the main text). As θ decreases, the two straight lines in the PBC spectrum approach each other and finally form a single closed loop, as given in Fig. S6. During this process, the width of the OBC spectrum keeps decreasing, making it within the loop of the PBC spectrum. In acoustic design, the imperfect coupling can be achieved by changing the number of tubes, as mentioned in Note 3.

Fig. S6 The eigen spectra for the PBC and OBC lattices derived from the transfer matrix method with **a**, $\theta=0.2\pi$, **b**, $\theta=0.4\pi$ and **c**, $\theta=0.47\pi$.

Reviewer comments:

For the missing technical details, how to design the acoustic ring resonators to obtain a near perfect coupling and hence the decoupled clockwise and anticlockwise modes over a large frequency regime is never discussed. This is very important, because otherwise coupled modes would ruin the condition for NHSE.

Response:

We appreciate this valuable suggestion. More details regarding the acoustic design have been added to the main text and supplementary information. Indeed, the clockwise and counterclockwise modes need to be decoupled to construct the pseudo-spin degree, and perfect coupling can help in the experimental observation of NHSE. However, perfect transmission is not a necessary condition for NHSE. The coupling region we employed to connect the rings is a four-port system. Thus, even if the coupling is imperfect (e.g., only four tubes), the pure circulation of acoustic energy in the site rings (acoustic energy circulates in the same site ring) is still guaranteed. We have accordingly created a new section in the Supplemental Information to describe the acoustic model design procedure, which reads

There are two main considerations in designing the acoustic structure. The first one is that we want to achieve perfect coupling (i.e., full transmission) between two coupled ring resonators. This is for the purpose of clear visualization of the NHSE (the propagation path can be easily visualized in the perfect coupling case). The second consideration is the minimization of the reflection during the coupling process. This is crucial since our theory is based on the assumption that the clockwise and counterclockwise modes are decoupled.

To obtain optimal structural parameters that fulfill above requirements, we consider the structure shown in the top panel of Fig. S4a, which consists of two acoustic waveguides coupled by small tubes with fixed spacings (0.47 cm) and widths (0.7 cm). First, we sweep the thickness of the tubes (denoted by b) and frequency for a structure with 16 tubes to determine the value of parameter b and working frequency range with near zero reflection. According to the numerical results shown in the bottom panel of Fig. S4b, we adopt $b=0.35$ cm and working frequency around 8250 Hz (denoted by the grey star in Fig. S4a). Then, the coupling strength can be varied by changing the number of tubes in the coupling region. As shown in Fig. S4b, 16 tubes achieve perfect coupling in the frequency range of our interest, compared to the cases with 4 tubes and 10 tubes.

To check the performance of the design, we numerically calculated the pressure field at 8250 Hz of a basic unit cell to distinguish the modes with different circulations. As shown in Fig. S5, the mode injected from lower left side can be perfectly transmitted to the lower right side, without any noticeable reflection.

Fig. S4 **The scattering properties of the coupling region.** **a** The reflection property as a function of the structural parameter b . **b** The coupling angle with different numbers of tubes: 4 tubes, 10 tubes and 16 tubes.

Fig. S5 **Acoustic wave propagation through one unit cell.**

Reviewer comments:

How to experimentally implement a complex frequency, which to my knowledge is very counterintuitive, is not discussed either.

Response:

The complex-frequency excitation $\omega = \omega_0 + i\omega'$ can be realized using a time-varying signal. As mentioned in our response to previous comment, the real part of frequency ω_0 is determined by the structural parameters of the acoustic design. Furthermore, the imaginary part of frequency ω' can be treated as the attenuated or amplified term of the signal, e.g., $e^{-\omega't} e^{i(\omega_0 t - kx)}$. In experiments, we compose the signal with complex

frequency in the computer, then send it to a programmable signal generator to launch the speaker. Following the reviewer's suggestion, we have added a discussion on the generation of complex-frequency excitation in the Methods part that reads "The complex-frequency excitation is generated by employing a time-varying sinusoidal signal. The real part of the complex excitation is 8250 Hz, which is at the center of working frequency window of the acoustic lattice. To determine the imaginary part of the excitation, we conduct the measurement shown in Fig. 4c-d with the imaginary part of the excitation gradually increased from zero. When virtual gain and loss effects are respectively realized in the lower and upper parts of the link ring, the suitable value for the imaginary part of the excitation is then identified, which is $412.5i$ Hz. Finally, we prepare the time-dependent signal in advance, then sent it to the programmable signal generator to trigger the sound signal from the speaker.". Besides, we also added a new plot (Fig. 4b) to show the measured electric signal imposed on the speaker, as well as an additional sentence at line 222 that reads "The measured time-varying electric signal imposed on the loudspeaker is shown in Fig. 4b."

Reviewer comments:

The descriptions of the effect of virtual gain are not clear. For example, sentences like "For a propagating mode with frequency ...the actual decay of the mode" are very confusing.

Response:

We agree with the reviewer that the mentioned descriptions are not satisfying. The sentence has been rewritten as "For an excitation with complex frequency ω , at a fixed instant, the intensity (normalized to the input) will increase along the propagation direction, when the decay in excitation is faster than the actual decay of the mode in the lattice at the corresponding frequency.". We also double-checked the revised manuscript to ensure an unambiguous presentation.

Reviewer #3

Reviewer comments:

In the manuscript “Transient non-Hermitian skin effect” by Zhongming Gu, et al., the authors utilize the so-called virtual gain to reveal the non-Hermitian skin effect (NHSE). The NHSE effect is possible in non-Hermitian 1D or 2D systems and consists of the localization of many modes at edges. In non-Hermitian passive systems, the NHSE effect is hidden in the complex frequency plane. Authors show that exponentially decaying signals (frequency with complex imaginary part) enable excitation of these localized modes. The approach is discussed using the general toy tight-binding model, analyzed numerically in the acoustic domain and realized experimentally. The reported results of the work are rather curious and inspiring and can be useful for a broad range of applications. Overall, I believe that the paper is worth publication in Nature Communication after addressing the following comments:

Response:

We would like to thank the reviewer for the encouraging comments.

Reviewer comments:

1. It would be instructive to discuss the NHSE effect more and, in particular, how it differs from other edge modes (for example, photonic Tamm states) and why one needs gain to achieve this effect at the real frequency.

Response:

We appreciate the reviewer’s insightful suggestion. More discussions have been added to the revised introduction part. At line 20, we added one sentence to discuss the difference between the skin modes and other edge modes: “**The boundary-localized eigenmodes in NHSE, namely the skin modes, only exist in point-gapped systems that are intrinsically non-Hermitian and thus are fundamentally different from other boundary modes such as the Tamm states and topological edge states that can arise without the aid of non-Hermiticity.**” At line 46, one more sentence to explain why we needed gain to observe the NHSE: “**In such a case, instead of observing the localization of the skin modes, one observes that the excited field is localized around the source**”.

Reviewer comments:

2. The model in Fig 3(a) comprises loss in the coupling, whereas the model in Fig 2 implies that the hopping strengths are real-valued. How this difference affects the results?

Response:

The configuration in Fig. 3a is a typical way to achieve nonreciprocal (or asymmetric) hopping (i.e., the hopping in Fig. 2) in ring resonator lattice. It has been adopted in a few literatures, e.g., Refs. [44-45] in the revised version and also [Optics Express 29, 8462 (2021)] which contains detailed derivations how gain/loss in link ring can be mapped to nonreciprocal hopping. Intuitively, such a mapping can be understood by examining the wave paths. In Fig. 3a, when the wave goes from left to right, it passes through the upper half of the link ring, experiencing a larger decay. When the wave goes from right to left, it instead passes

through the lower half of the link ring, experiencing a smaller decay. Therefore, hopping between the two site rings connected by such a link ring is effective nonreciprocal.

Reviewer comments:

3. Please explain in the text how without the γ term, the eigenvalues of (1) under periodic boundary conditions (PBC) form a closed loop in the complex plane? I guess the Hamiltonian, in this case, is Hermitian and non-hermicity comes from the boundary conditions.

Response:

The γ term represents the on-site loss or background loss, which shifts the eigen-spectrum along the imaginary axis in the complex plane. Without the γ term, the Hamiltonian is a standard HN model (which is non-Hermitian regardless of the boundary condition), and the spectrum under PBC is still a closed loop in the complex plane. The non-Hermiticity when $\gamma=0$ comes from the nonreciprocal hopping (i.e., $\kappa_r \neq \kappa_l$). What a nonzero γ does is to render all modes under PBC lossy while maintaining the system's point gap topology. According to the suggestion of Reviewer #3, we have added a comment at line 102 that reads “When $\gamma=0$, this lattice is known to host NHSE induced by the asymmetric hoppings (i.e., $\kappa_r \neq \kappa_l$), with the eigen spectrum under periodic boundary condition (PBC) forms a closed loop in the complex plane and the eigenmodes under open boundary condition (OBC) are all skin modes. A nonzero γ just shifts the eigenvalues of all modes along the imaginary axis while keeping the nontrivial point gap topology unchanged” in the revised manuscript.

Reviewer comments:

4. Page 2: “the intensity will increase along the propagation direction when the decay in excitation is faster than the actual decay of the mode.” Actually, not intensity, but the ratio of it to the incoming signal would increase in time.

Response:

We thank the reviewer for pointing this out. We have changed this sentence as “For an excitation with complex frequency ω , at a fixed instant, the intensity (normalized to the input) will increase along the propagation direction when the decay in excitation is faster than the actual decay of the mode in the lattice at the corresponding frequency.” in the revised manuscript.

Reviewer comments:

5. What is ϕ in Fig 3(c)? Does it relate to the frequency?

Response:

The physical meaning of ϕ is the accumulated phase of acoustic wave over one round trip in one ring. It is directly related to the frequency. Besides, it serves as quasi-energy in the ring resonator lattice model. We have rewritten the sentence to clarify the meaning of ϕ that reads “Given the boundary condition, the scattering equations can be cast into a Floquet eigen-problem, with the round trip phase ϕ in each ring playing the role of quasi-energy (see Supplementary Note 1 for more details).”

Reviewer comments:

6. Please explain why the coupling regions' scattering matrix (2) is unitary, given that the couplers are lossy?

Response:

In our ring resonator lattice model, the lossy regions are not included in the coupling region. To clarify this point, we have added two dashed boxes in Fig. 3a in the main text to denote the coupling regions, as well as one extra sentence at line 158 that reads “...denoted by the red dashed boxes in Fig. 3a”. It can be observed that the lossy elements, which are located at the upper and lower half of the link ring, do not fall within the coupling regions. Thus, the scattering matrix is taken to be unitary.

Reviewer comments:

7. The concept of virtual gain was also proposed in the concept of optical force control [S Lepeshov, A Krasnok, Virtual optical pulling force, Optica 7 (8), 1024-1030 (2020)]

Response:

We thank the reviewer for bringing this related work to us. We have cited this paper in the revised manuscript as Ref. [41] and added a comment that reads “This technique has also been applied to achieve virtual optical pulling force.⁴¹”.

REVIEWERS' COMMENTS

Reviewer #1 (Remarks to the Author):

The authors have addressed my comments and improved the manuscript significantly. I am happy to recommend publishing this work in Nature Communications.

Reviewer #2 (Remarks to the Author):

The authors have made considerable revisions to the manuscript and included necessary details that were lacked in the previous manuscript, especially on the justifications of the novelty and specificities of the implementation of the complex frequency. With their revisions, now I have been convinced and recommend publication.

Reviewer #3 (Remarks to the Author):

I thank the Authors for addressing my comments as well as the comments of other referees. Now I can recommend the paper for publication.